# Syndecan-3 Coregulates Milk Fat Metabolism and Inflammatory Reactions in Bovine Mammary Epithelial Cells through AMPK/SIRT1 Signaling Pathway

**DOI:** 10.3390/ijms24076657

**Published:** 2023-04-03

**Authors:** Jing Fan, Zhihui Zhao, Haochen Wu, Xibi Fang, Fengshuai Miao, Xuanxu Chen, Xinyi Jiang, Jing Li, Ping Jiang, Haibin Yu

**Affiliations:** 1College of Coastal Agricultural Sciences, Guangdong Ocean University, Zhanjiang 524088, China; fj13359819865@163.com (J.F.); zhzhao@gdou.edu.cn (Z.Z.); 13275125309@163.com (H.W.); miaofengshuai@163.com (F.M.); chenxuanxugdou@163.com (X.C.); jxy991130@163.com (X.J.); 18770421250@163.com (J.L.); 2The Key Laboratory of Animal Resources and Breed Innovation in Western Guangdong Province, Zhanjiang 524088, China; 3College of Animal Science, Jilin University, Changchun 130062, China; fangxibi@jlu.edu.cn

**Keywords:** SDC3, bovine mammary epithelial cells, AMPK/SIRT1, milk fat metabolism, inflammatory reaction

## Abstract

Transcriptome sequencing showed that syndecan-3 (*SDC3*) was differentially expressed in high-fat and low-fat mammary epithelial cells of Chinese Holstein cows. Previous studies found that *SDC3* plays an important role in inflammatory diseases and virus infection. However, those studies did not confirm whether or not the functional gene *SDC3*, which plays an important role in regulating milk fat metabolism, has an effect on susceptibility to breast tissue diseases. Therefore, we studied the effects of *SDC3* on milk lipid metabolism and inflammation in bovine mammary epithelial cells (BMECs) and further explored the common regulatory pathway of *SDC3* in both. The overexpression of *SDC3* increased the contents of triglycerides and cholesterol, reduced the content of non-esterified fatty acids, inhibited the expression of inflammatory factors (*IL-6*, *IL-1β*, *TNF-α* and *COX-2*), and reduced the production of ROS in BMECs. However, silenced *SDC3* had the opposite effect. Further exploring the mechanisms of *SDC3*, we found that *SDC3* upregulated the expression of peroxisome proliferator-activated receptor gamma (*PPARG*) through the *AMPK/SIRT1* signal pathway to promote milk fat synthesis. It also regulated the activation of the *NF-κB* pathway through the *AMPK/SIRT1* signal pathway, reducing the expression of inflammatory factors and ROS production, thus inhibiting the inflammatory response of BMECs. Nuclear factor kappa B subunit 1 (*NF-κB p50*) was an important target of *SDC3* in this process. To sum up, our results showed that *SDC3* coregulated milk fat metabolism and inflammation through the *AMPK/SIRT1* signaling pathway. This study laid a foundation for the comprehensive evaluation of breeding value based on multi-effect functional genes in dairy cow molecular breeding.

## 1. Introduction

Milk fat is one of the main substances that make up milk nutrition. However, mastitis, a very common disease in dairy cows, can lead to a decline in breast lactation function and reduce milk quality. Some studies found that the contents of long-chain fat (C16-C18), total fatty acids [1], polyunsaturated fatty acids [2], and milk fat in milk from recessive mastitis cows were lower than in milk from healthy cows, indicating a close relationship between dairy quality traits and breast inflammation. Our team used high-fat and low-fat BMEC lines for transcriptome analysis, without the interference of immune cells. In addition to obtaining a large number of functional milk-fat-metabolism-related genes, it was determined that some immune-related genes are differentially expressed in high-fat and low-fat bovine mammary epithelial cells (BMECs) [3]. Therefore, we speculated that there could be a coordinate regulation network between fat metabolism and immune traits that could jointly regulate milk quality traits and mastitis susceptibility. At the same time, this phenomenon also suggested that some genes with complex biological functions could coordinate the regulation on milk fat metabolism and immune traits. However, whether the complexity of the biological functions of these genes affects breeding value evaluations and their application in the breeding process remains unknown. Thus, determining whether the comprehensive effects of functional genes need to be considered in the future molecular breeding of functional genes (which play an important role in regulating milk fat metabolism) could be worthwhile. What is the coordinate regulation of fat metabolism-related genes in BMECs and immunomodulatory factors? Do functional genes that play an important role in regulating milk fat metabolism affect the susceptibility to breast tissue diseases? These questions need to be answered through the functional verification of candidate genes.

The heparan sulfate proteoglycan (syndecan, SDC) gene family is a type 1 transmembrane protein. It is composed of core proteoglycan and its covalent glycosaminoglycan polypeptide chain. Its core protein is composed of a variable length amino terminal, a transmembrane region, and a short conserved cytoplasmic region. The SDC family consists of four cell surface molecules (*SDC1* to *SDC4*) with different biological functions, of which syndecan-3 (*SDC3*) is the largest. It was first cloned in the rat from a Schwann cell and a rat brain library [4]. The full-length 38589 bp of bovine *SDC3* gene, located on chromosome 2, consists of 5 exons and 4 introns, encoding a total of 346 amino acids. As a new regulator of feeding behavior and body weight, *SDC3* is highly expressed in the hypothalamus, which controls energy balance [5], and participates in the regulation of melanocortin system activity [6], effectively regulating energy intake, energy consumption, and peripheral glucose metabolism. In addition to participating in energy metabolism as a weight regulator, *SDC3* is also directly related to fat metabolism. It was found that mice with *SDC3* gene deletion (induced by high-fat diet) showed resistance to obesity, significantly reduced fat accumulation, and good glucose tolerance [7]. Two single nucleotide polymorphisms in exon 3 of *SDC3* were significantly associated with obesity [8]. Based on the regulatory role of *SDC3* in fat metabolism, it was speculated that the gene could be involved in milk fat metabolism. The transcriptome research results of our group showed that *SDC3* showed differential expression in the transcriptome data analysis of high-fat and low-fat mammary gland epithelial cells of Chinese Holstein cows [3]. This suggested that our *SDC3* gene could be involved in—and regulate—milk fat metabolism. Furthermore, *SDC3* plays a dual role in inflammation. *SDC3* showed pro-inflammatory effects in joint tissue, and leukocyte recruitment and cartilage damage in antigen-induced arthritis mice were significantly lower than those in wild-type mice due to *SDC3* gene deletion. However, *SDC3* had an anti-inflammatory effect in skin and secretions, and leukocyte rolling and adhesion increased in skin and genital inflammation models, indicating that *SDC3* had anti-inflammatory activity [9]. For cells, *SDC3* is expressed in many cell types, participates in the regulation of growth factor signal [10], cell adhesion and migration [11,12], and proliferation and differentiation [13], and also plays an important role in inflammation and virus infection [14]. These results suggest that the *SDC3* gene not only participates in milk fat metabolism, but also plays an important role in immune regulation. The effects of the *SDC3* gene on milk fat metabolism and inflammation, and how the gene plays a coregulatory role in milk fat traits and immune characteristics, are worthy of further study. Analysis of the function of *SDC3* gene could contribute to the comprehensive selection of related traits in the breeding of milk fat and immune traits in dairy cows in the future.

*AMPK* is a serine/threonine protein kinase that plays an important role in regulating energy homeostasis [15], cell growth [16], and lipid metabolism [17]. It was further found that *AMPK* was a key factor for trans-10, cis-12 conjugated linoleic acid (t10c12-CLA) to regulate milk fat synthesis in goat mammary epithelial cells [18]. *AMPK* protein kinase is a heterotrimer consisting of one catalytic subunit (α) and two regulatory subunits (β and γ) [19]. Protein kinase AMP-activated catalytic subunit alpha 1 (*AMPKα1*) is mainly expressed in breast tissue [20], and the phosphorylation of α-subunit Thr172 site is a necessary condition for the full activation of *AMPK*. In addition, *AMPK* can activate sirtuin 1 (*SIRT1*) [21], which regulates glucose and lipid metabolism and tumorigenesis, whereas *SIRT1* can regulate the core factor of fat synthesis, peroxisome proliferator activated receptor gamma (*PPARG*) [22], and the key regulator of the inflammatory network, *NF-κB* [23]. *NF-κB* is the aggregation point of many signal transduction pathways and plays an important role in regulating inflammation, immune response, and stress response [24]. The phosphorylation of its inhibitory protein, *IκB*, is a necessary condition for its release and entry into the nucleus before it can exert its biological function [25].

Based on previous studies and the results of our transcriptome, we knew that *SDC3* participated in and regulated milk fat metabolism and inflammation, but the effects of *SDC3* on milk fat metabolism and inflammation and its molecular mechanism in BMECs had not yet been reported. Furthermore, according to the above description, it was interesting that the *AMPK* signal pathway regulated milk fat synthesis, whereas *NF-κB* was a key regulator in the inflammatory network, involved in the regulation of inflammatory response. We wanted to know whether *SDC3* coregulated milk lipid metabolism and inflammation of BMECs through these two signal pathways. Therefore, in this study, we aimed to explore the functions of *SDC3* in milk fat metabolism and inflammation, and to further analyze the coregulation pathway of *SDC3* on milk lipid metabolism and immune traits in BMECs.

## 2. Results

### 2.1. Identification of Transfection Efficiency of Plasmid and siRNAs in BMECs

To ensure the effectiveness of the *SDC3* overexpression vector and inhibitor siRNA in BMECs, after the vectors and siRNAs were transfected into cells, green fluorescence was examined using fluorescence microscopy. RT-qPCR and Western blot were also used. After 24 h, green fluorescence was observed under a fluorescence microscope. Findings suggested that the vectors (pBI-CMV3-SDC3 and pBI-CMV3) and siRNAs were successfully transfected into the BMECs (Figure 1A,E). Compared with the control, the overexpression of *SDC3* significantly increased the levels of *SDC3* transcription (*p* < 0.01) (Figure 1B) and protein (*p* < 0.01) (Figure 1C,D). The transcriptional levels of *SDC3* in BMECs transfected with siRNA-1218, siRNA-791 and siRNA-979 decreased by 70%, 57%, and 6%, respectively (Figure 1F), and siRNA-1218 significantly decreased the levels of *SDC3* protein (*p* < 0.05) (Figure 1G,H). siRNA-1218 had the highest efficiency of *SDC3* knockdown and was used as an inhibitor of *SDC3* in subsequent experiments.

### 2.2. SDC3 Promotes the Synthesis of Milk Fat in BMECs

To study the effects of *SDC3* on milk lipid metabolism in BMECs, the BMECs were seeded on 6-well plates and harvested after 24 h, post-transfection with vectors or siRNAs. The contents of triglycerides, cholesterol, NEFA, and the distribution of lipid droplets in BMECs were used to evaluate the effects of *SDC3* on milk fat metabolism in BMECs. The results showed that, compared with the control group, the overexpression of *SDC3* significantly increased the content of triglycerides (*p* < 0.05) (Figure 2A) and cholesterol (*p* < 0.01) (Figure 2B) but significantly decreased the NEFA content (*p* < 0.01) (Figure 2C). On the contrary, compared with the negative control group, the siRNA-1218 group obviously decreased the contents of triglycerides (*p* < 0.05) (Figure 2D) and cholesterol (*p* < 0.01) (Figure 2E). They also decreased the accumulation of lipid droplets (Figure 2G) but obviously increased NEFA content (*p* < 0.05) (Figure 2F). To sum up, the overexpression of *SDC3* was able to significantly promote the synthesis of triglycerides and cholesterol, which are the main components of milk fat in BMECs. It also played a positive role in regulating milk fat synthesis while promoting the metabolism of NEFA. SDC3 knockout had the opposite effect.

### 2.3. SDC3 Inhibits the Inflammatory Response of BMECs by Inhibiting the Expression of Inflammatory Factors and the Production of ROS

In order to investigate whether *SDC3* was able to regulate the inflammatory response in BMECs, we tested the effects of overexpression of *SDC3* and silencing *SDC3* interventions on the levels of inflammatory markers (*IL-6*, *IL-1β*, *TNF-α* and *COX-2*) in normal growth BMECs at both the transcriptional and translational level. This was carried out using RT-qPCR and ELISA assays, whereas intracellular ROS generation detection was accomplished using flow cytometry. The results of our RT-qPCR assay were consistent with the changes observed in the ELISA assays, which showed that the expression levels of interleukin 6 (*IL-6*), interleukin 1 beta (*IL-1β*), and tumor necrosis factor (*TNF-α*) in cells were significantly downregulated in the overexpression of *SDC3* group versus the control group (*p* < 0.05) (Figure 3A–F). On the other hand, the expression levels of these inflammatory factors in cells were significantly upregulated in the silencing *SDC3* group versus the control group (*p* < 0.05) (Figure 3G–L). The results of the flow cytometry showed that the overexpression of *SDC3* could significantly inhibit (*p* < 0.01) intracellular ROS generation, whereas knockout of *SDC3* could increase (*p* < 0.01) the levels of intracellular ROS (Figure 3O–R). At the same time, the transcriptional levels of cytochrome c oxidase subunit II (*COX-2*), which plays a central role in ROS generation [26], decreased with the overexpression of *SDC3* and increased when *SDC3* was silenced (*p* < 0.05) (Figure 3D,N). In short, the overexpression of *SDC3* without inflammatory stimulation decreased the expression of inflammatory factors and ROS generation in BMECs and inhibited the occurrence of inflammatory responses to a certain extent. Thus, *SDC3* demonstrated anti-inflammatory effects and could potentially regulate intracellular inflammatory responses.

### 2.4. SDC3 Exerts a Cooperatively Regulatory Effect on Milk Lipid Metabolism and Inflammatory Reactions in BMECs through the AMPK/SIRT1 Signal Pathway

#### 2.4.1. SDC3 Promotes Milk Fat Synthesis through AMPK/SIRT1/PPARG Signal Pathway

Previous studies have shown that the *AMPK* signaling pathway plays an important role in fat synthesis [17]. *SIRT1*, known as NAD-dependent deacetylase, can be activated by *AMPK* to inhibit lipid accumulation [21]. It can also inhibit the expression of *PPARG*, the core regulator of fat synthesis, to reduce fat synthesis [22]. Therefore, we transfected the vector (pBI-CMV3-SDC3 and pBI-CMV3) and siRNAs (siRNA-1218 and siRNA-NC) into BMECs. We then detected the effects of *SDC3* on *AMPK* and its downstream genes using RT-qPCR and Western blot methods in order to explore the mechanism by which *SDC3* promoted milk fat synthesis. The results showed that with pBI-CMV3 as the negative control, the overexpression of *SDC3* inhibited activation of the *AMPK* signal pathway by significantly reducing the protein expression of *p-AMPK* (*p* < 0.05) (Figure 4C,D) and the transcriptional level of *AMPK* (*p* < 0.01) (Figure 4A). It also decreased the expression of its downstream gene SIRT1 (*p* < 0.01) (Figure 4A,C,E) and increased the expression of PPARG (*p* < 0.05) (Figure 4A,C,E). On the contrary, silencing SDC3 significantly promoted the protein expression of *p-AMPK* (*p* < 0.05) (Figure 4F,G), had no effect on transcriptional levels of *AMPK* (Figure 4B), and increased the expression of SIRT1 (*p* < 0.05) (Figure 4B,F,H). It also activated the *AMPK/SIRT1* signal pathway, thus inhibiting the expression of *PPARG* (*p* < 0.05) (Figure 4B,F,H). These results suggested that SDC3 upregulated the expression of *PPARG*, the core regulator of fat synthesis, by inhibiting the *AMPK/SIRT1* signal pathway—which led to the increase in fat synthesis and accumulation, promoting milk fat synthesis.

#### 2.4.2. SDC3 Inhibits Inflammatory Reaction through the AMPK/SIRT1/NF-κB P50 Signaling Pathway

The above results showed that *SDC3* significantly inhibited the expression of inflammatory factors and ROS generation, and had a good anti-inflammatory effect, as well. Thus, it was necessary for us to further explore the anti-inflammatory mechanism of *SDC3*. Because *NF-κB* can activate the transcription of inflammatory cytokines *IL-6*, *IL-1β*, *TNF-α*, and *COX-2* [27], and *AMPK* and *SIRT1* participate in the regulation of inflammatory network as upstream regulators of *NF-κB* [23], we studied the effects of *SDC3* on *NF-κB*, a key factor in the inflammatory network. The overexpression of *SDC3* significantly inhibited the expression of *NF-κB p50* (*p* < 0.01) (Figure 5A,C,D), decreased the protein expression of *P-IκBα* (*p* < 0.05) (Figure 5C,D), and increased the expression of inhibitory protein *IκBα* (*p* < 0.01) (Figure 5C,D). This inhibited the migration of *NF-κB p50* to the nucleus and the release of downstream inflammation-related transcription factors (*IL-6*, *IL-1β*, *TNF-α* and *COX-2*), thereby also inhibiting the inflammatory responses of BMECs. However, in the absence of inflammatory stimulation, with siRNA-NC as the negative control, silencing SDC3 significantly increased the expression of *NF-κB p50* (*p* < 0.01) (Figure 5B,E,F) and upregulated the protein expression of *P-IκBα* (*p* < 0.05) (Figure 5E,F). Thus, the inhibitory protein *IκBα* was ubiquitinated and degraded (*p* < 0.05) (Figure 5E,F), and *NF-κB p50* entered the nucleus to activate downstream transcription factors, promoting inflammatory responses, increasing intracellular ROS generation, and resulting in cell damage. The results suggested that *NF-κB p50* could be an important target for *SDC3* to regulate *NF-κB* pathways, as *SDC3* activated or inhibited the *NF-κB* signal pathway by regulating its expression, and then regulated the inflammatory response of BMECs. In summary, *SDC3* regulated the expression of inflammatory cytokines *IL-6*, *IL-1β*, *TNF-α*, and *COX-2* and the levels of intracellular ROS through the *AMPK/SIRT1/NF-κB p50* signal pathway in order to regulate the inflammatory responses of BMECs (Figure 6).

## 3. Discussion

BMECs are the basic unit of synthesizing and secreting milk and are the target cells for studying the mechanism of lactation in breast tissue in vitro. In this study, BMECs were used as the research object to explore the effects of *SDC3* on milk fat synthesis by transfection of siRNAs and plasmids in vitro. The main components of milk fat include 97–98% triglycerides, 0.2–0.4% cholesterol, and 0.1% fatty acids [28]. The overexpression of *SDC3* led to increases in triglycerides and cholesterol content and decreases of NEFA content in BMECs. On the contrary, silencing the *SDC3* gene decreased triglyceride content, cholesterol content, and lipid droplet accumulation in BMECs, and increased NEFA content. It was suggested that *SDC3* could regulate milk fat metabolism in BMECs. *AMPK* is a recognized protein that participates in various life activities by regulating energy balance [15]; it has been significantly associated with obesity and lipid metabolism disorders. Therefore, we detected the expression levels of *p-AMPK* after overexpression and silencing of *SDC3* to explore whether *SDC3* regulated milk fat metabolism by activating the *AMPK* signal pathway. We found that the overexpression of *SDC3* decreased the phosphorylation level of *AMPK* and significantly inhibited the activation of *AMPK*. Silencing *SDC3* significantly increased the phosphorylation levels of *AMPK* and activated the *AMPK* signal pathway, suggesting that *SDC3* was able to regulate milk fat metabolism through that pathway. In addition, the activation of *AMPK* increased the intracellular NAD+ concentration, and then activated *SIRT1*, which jointly regulated energy consumption [21,29]. It has been observed that LB100 attenuates FFA-induced lipid accumulation in normal human hepatic cell lines through the *AMPK/SIRT1* signal pathway [30]. *SIRT1* is a NAD+-dependent protein deacetylase that regulates the activity of downstream signal proteins by deacetylating these proteins. It is widely involved in cell senescence, glucose and lipid metabolism [31], and other life processes. Crucially, *SIRT1* inhibits fat production and promotes fat mobilization by inhibiting *PPARG* [22], which plays a central role in adipocyte differentiation [32] and fat deposition [33]. We found that the overexpression of *SDC3* could significantly reduce the expression of *SIRT1* and increase the expression of *PPARG*. However, the opposite was observed after silencing *SDC3*, which indicated that *SDC3* was involved in milk fat synthesis as a positive regulator. Previous studies have found that both *p-AMPK* and *SIRT1* cooperatively inhibited the expression of *PPARG*, resulting in the inhibition of lipid synthesis [34]. Gynostemma Pentaphyllum Extract (GPE) decreased the levels of serum triglycerides and total cholesterol in mice fed a high-fat diet and inhibited increased body weight, fat mass, and adipocyte hypertrophy through the *AMPK/SIRT1/PPARG* signaling pathways [35]. In summation, consistent with previous studies, *SDC3* was shown to regulate milk fat synthesis in BMECs through the *AMPK/SIRT1/PPARG* signal pathways.

BMECs have an important physiological function: synthesizing and secreting milk. However, they also play an important role in the process of antimicrobial invasion of mammary glands. When external pathogenic microorganisms invade breast tissue, BMECs make first contact with pathogenic microorganisms. They can identify pathogens and play a sentinel role [36], which could be of great significance to the development and prognosis of dairy cow mastitis. Studies showed that when breast tissue was infected, BMECs were able to synthesize and secrete a variety of cytokines, such as *TNF-a, IL-lβ*, and *IL-6* [37]. *E. coli* produced a strong, rapid inflammatory response, inducing BMECs to produce *TNF-α* and *IL-1β*. The inflammatory response of BMECs induced by S. aureus mainly depended on the production of *IL-6* [38]. In addition, it is well known that excessive production of ROS can cause inflammation and cell damage, or even cell death, by inducing apoptosis. The results of RT-qPCR, ELISA, and flow cytometry showed that the overexpression of *SDC3* could notably downregulate expression of *IL-6, IL-1β, TNF-α*, and *COX-2* and reduce ROS generation in BMECs without inflammatory stimulation. On the other hand, silencing *SDC3* led to the high expression of *IL-6, IL-1β, TNF-α*, and *COX-2* and increased the level of intracellular ROS. This indicated that *SDC3* was able to regulate the inflammatory response of BMECs and exert certain anti-inflammatory effects. This result was consistent with the findings of Kehoe et al., who found that, in the model of skin inflammation and secretory inflammation, the absence of *SDC3* enhanced recruitment and the interaction between leukocytes and endothelial cells [39]. The rolling and adhesion of leukocytes in *SDC3*-deficient mice increased, suggesting that *SDC3* played an important role in anti-inflammation, as well [40]. In addition, because both the transmembrane and cytoplasmic regions of the SDC family core proteins have binding sites for protein kinase phosphorylation and multiple signal molecules [41], SDC can be used as receptors for cytokines, chemokine, and growth factors to regulate different signal pathways [42,43]. We found that inflammatory markers (*IL-6, IL-1β, TNF-α*, and *COX-2*) regulated by *SDC3* were downstream genes of the key *NF-κB* signaling pathway in the inflammatory response [44]. Therefore, we speculated that *SDC3* could regulate the inflammatory response of BMECs through the *NF-κB* signal pathway. 

Transcription factor kappa B was first discovered in 1986. It was so named because *NF-κB* binds to the enhancer element of the immunoglobulin κ light chain and then activates B cells [45]. In resting cells, *NF-κB* forms a complex with its inhibitory protein IκB, masking the nuclear localization signal of *NF-κB*, which results in it remaining in the cytoplasm as an inactive complex. When stimulated, *IκBα* is phosphorylated by its kinase IKK and degraded, and the activated *NF-κB* is transferred to the nucleus to bind to the KB site of the target gene promoter, thus playing the role of transcriptional regulation [25]. Our study found that the overexpression of *SDC3* significantly decreased the phosphorylation levels of *IκBα*, increased the expression levels of *NF-κB* inhibitor protein *IκBα*, and then inhibited the activation of *NF-κB*. On the contrary, silencing *SDC3* significantly increased the phosphorylation level of *IκBα*, degraded *IκBα*, and released *NF-κB* into the nucleus to activate downstream transcription factors, thereby increasing the expression of inflammatory factors and intracellular ROS levels. These results demonstrated that *SDC3* regulated the inflammatory response of BMECs through the *NF-κB* signaling pathway. It is well known that *NF-κB* mainly exists as a heterodimer, composed of *NF-κB p65*, and *NF-κB p50* subunits. Therein, *NF-κB* p65 shows transactivation activity through the transactivation domain at the C-terminal, whereas *NF-κB p50* directly binds to a specific DNA sequence (the *NF-κB* binding element). These two subunits jointly target gene promoters in various inflammatory responses. Previous studies showed that SIRT1 deacetylated *NF-κB p65* at lysine 310, thereby reducing the transcriptional activation of *NF-κB* and inhibiting the inflammatory response mediated by *NF-κB* [27]. However, our results showed that silencing *SDC3* could increase the phosphorylation of *AMPK* and upregulate the expression of downstream gene SIRT1. The upregulation of SIRT1 expression did not inhibit the activation of *NF-κB*, however. On the contrary, *NF-κB* was activated. Therefore, we explored the effects of *NF-κB p50*. The results showed that, after silencing *SDC3*, the expression of *NF-κB p50* increased and then activated the *NF-κB* pathway, whereas the overexpression group of *SDC3* reacted to the contrary. Consistent with our studies, Eriocalyxin B (ERIB) interfered with the binding of *NF-κB* and response elements by targeting *NF-κB p50*, thus exerting its anti-autoimmune inflammatory activity [46]. In Mesangial cells (MCs), LincRNA-Gm4419 can activate the *NF-κB* pathway through interaction with the *NF-κB* subunit p50 [47]. All in all, from our results, we were able to conclude that *SDC3* regulated the activation of the *NF-κB* signaling pathway through *NF-κB p50*, thus exerting its anti-inflammatory activity.

## 4. Materials and Methods

### 4.1. Reagents and Antibodies

DMEM and DPBS were purchased from HyClone (UT, USA). Fetal bovine serum (FBS) was purchased from ZATA Life (CA, USA). The triglycerides and cholesterol detection kit were purchased from Pulilai (Beijing, China). The non-esterified fatty acids assay kit was purchased from Nanjing Jiancheng Bioengineering Institute (Nanjing, China). The lipid dye BODIPY505/515 was purchased from Invitrogen (Waltham, MA, USA), and three ELISA kits (*IL-6*, *IL-1β* and *TNF-α*) were purchased from MEIMIAN (Wuxi, China). The primary antibodies against *AMPKα1*, Phosphorylated AMP-activated protein kinase (*p-AMPK*), *SDC3*, *PPARG*, and Phospho-IκB Alpha (*p-IκBα*) were purchased from Bioss (Beijing, China). Antibodies against nuclear factor kappa B subunit 1 (*NF-κB p50*), *IκBα*, and *SIRT1* were purchased from Proteintech Group (Wuhan, China). The antibodies against *β-actin* and anti-rabbit secondary antibodies were acquired from Bioworld Technology (St. Louis Park, MN, USA).

### 4.2. Cell Culture

The BMECs used in this study were derived from the mammary gland tissue of Chinese Holstein cows and were provided by the Laboratory of Molecular Genetics, College of Animal Science, Jilin University [48,49]. Throughout the experiment, BMECs were cultured in DMEM-F12 with 10% fetal bovine serum at 37 °C, in humidified air with 5% CO_2_. When the fusion degree of BMECs in the six-well plates (Nest Biotechnology, Shanghai, China) reached 80%, the plasmid or small interfering RNA (siRNA) was transfected into the cells. After 24 h, the overexpression and inhibition efficiency of cells were detected by observation of green fluorescence under microscope, RT-qPCR, and Western blotting methods.

### 4.3. Transfection

Plasmids with pBI-CMV3-SDC3 vectors containing SDC3 overexpression sequences were given by Jilin University. Plasmids with pBI-CMV3 vectors containing any non-target sequences were used as negative controls. The SDC3 siRNA pool and its negative control, siRNA-NC, were designed and manufactured by GenePharma Company (GenePharma, Shanghai, China). The SDC3 siRNA pool contained three siRNAs targeting different portions of the SDC3 mRNA sequence. They were designed to have no homology with any bovine gene. The sequences were: siRNA-1218, 5′-GCUCAUCUACCGCAUGAAGTT-3′; siRNA-791, 5′-CCACCCAGAAGCCUGAUAUTT-3′; siRNA-979, 5′-CCAGACACAGCCAAUGAAGTT-3′; and siRNA-NC, 5′-UUCUCCGAACGUGUCACGUTT-3′. In short, 1 × 10^6^ cells were inoculated into a 6-well plate until the fusion reached about 80%. Then, 3μg of vectors were transfected into the cells using 6 μL of ViaFect (Promega, WI, USA) and 9.4μL of siRNAs were transfected into the cells using 3.75 μL of lipo3000 (Invitrogen, MA, USA) in a serum-free medium for 6 h. Then, the medium was supplemented with serum and maintained in culture for 24 h. After that, the transfection efficiency was observed using fluorescence microscopy (ESCO, Singapore), and the cells were collected for subsequent experiments.

### 4.4. Measurement of Triglycerides and Cholesterol

After the 24 h transfection, BMECs were lysed and collected, and the concentrations of standards, triglycerides, and cholesterol were measured using a triglyceride kit and a cholesterol kit. In addition, a BCA protein assay kit (TaKaRa, Beijing, China) was used to measure the protein concentration in the cells. A standard curve was plotted to calculate triglyceride and cholesterol concentrations. The triglyceride/cholesterol to protein concentration ratio represented the final utilized data.

### 4.5. Non-Esterified Fatty Acids (NEFA) Measurement

A full 24 h after the plasmid or siRNAs were transferred into the BMECs, the content of NEFAs in the cell was detected using a non-esterified fatty acid kit. Briefly, after the cells were lysed with lysate, the cells were collected and bathed in water at 70 °C for 10 min and then centrifuged. A total of 4 µL of supernatant was placed in 96-well plates and mixed with 200 µL of a working solution, then cultured at 37 °C in the dark for 5 min. A microplate reader measured the absorbance as A1. Then, 50 µL of B working solution was added and incubated under the same conditions and the absorbance was measured to be A2. Protein concentration was detected using a BCA protein assay kit. Both absorbance values were measured at 546 nm, and the following formula calculated the content of NEFA in the cells:
$$\begin{array}{c} \;\;\mathrm{Intracellular}\;\mathrm{NEFA}\;\mathrm{concentration}(\mathrm{mmol}/\mathrm{gprot})\\ \;\;=\{(A1\;-\;A2)\;\mathrm{sample}\;-\;(A1\;-\;A2)\mathrm{blank}\}/\{(A1\;-\;A2)\;\mathrm{standard}\;-\;(A1\;-\;A2)\mathrm{blank}\}\;*\\ \mathrm{Standard}\;\mathrm{concentration}\;(\mathrm{mmol}/L)/\mathrm{Intracellular}\;\mathrm{protein}\;\mathrm{concentration}\;(\mathrm{gprot}/L) \end{array}$$

### 4.6. BODIPY Staining

siRNA-1218 and siRNA-NC were transfected into the cells by lipo3000 for 24 h. Then, they were washed with precooled PBS three times. Then, lipid dye BODIPY505/515 was added, diluted by DMEM, and stained in the dark for 30 min, before an additional washing with PBS (three times). Then, the cells were fixed with a fixation solution (Biosharp, Anhui, China) for 5 min and washed an additional three times with PBS for 5 min each. DAPI (Beyotime, Shanghai, China), diluted by PBS, was added and stained in the dark for 10 min. The cells were washed with PBS three times for 5 min, and the distribution of lipid droplets was observed under a laser confocal microscope (Olympus, Tokyo, Japan).

### 4.7. Protein Expression Analysis

The protein was extracted from the transfected cells and the protein concentration was measured by BCA protein assay kit (TaKaRa, Beijing, China). The total protein concentration of the control and treatment groups was standardized by adjusting the volume of the protein lysate. Sample diluent (40 μL), protein of cells (10 μL), and standard sample (50 mL) were added to appropriate wells. In addition to the blank control wells, enzyme-labeled reagents (100 μL) were added to the other wells and incubated at 37 ℃ for 1 h. The plate was washed with wash buffer five times. After drying, Substrate Solution A and Substrate Solution B (50 μL) were added to each well for 15 min of incubation. Finally, Stop Solution (50 μL) was added into each well and they were placed into a microplate reader (BioTek, VT, USA) to detect the absorbance value under 450 nm.

### 4.8. Cell ROS Analysis

After treatment according to the requirements of the transfection test, intracellular reactive oxygen species (ROS) production was detected using a 2′,7′-dichlorofluorescin diacetate (DCFH-DA, Beyotime, Shanghai, China) probe. Then, the treated cells were collected, after pancreatin digestion, and co-incubated in 37 °C, in darkness, with serum-free culture medium containing 5 μM DCFH-DA. After 20 min, they were washed three times with serum-free medium. Finally, the cells were centrifuged and resuspended in 500 μL PBS. A flow cytometer (Beckman Coulter, Brea, CA, USA) was used to analyze ROS levels in BMECs.

### 4.9. Gene Expression Analysis

According to the steps of the RNA extraction kit (TaKaRa, Beijing, China), the cells were lysed in 6-well plates with buffer RL lysate containing 2% DTT in order to extract total RNA. The OD value and RNA concentration were evaluated using a Nanodrop Lite (Thermo Scientific, Waltham, MA, USA) spectrophotometer. A reverse transcription kit (TaKaRa, Beijing, China) was applied to reverse-transcribe the 1μg RNA into cDNA. RT-qPCR primers (Table 1) were designed by primer5 software and analyzed by BLAST to confirm the specificity of each pair of primers to its target gene. Then, the cDNA was mixed with the primers of the target gene (Table 1) and the Chamq Universal SYBR qPCR master mix (Vazyme, Nanjing, China) was used for amplification reaction in a real-time PCR detection system (BIO-RAD, Hercules, CA, USA). β-actin was used as the internal reference gene. Each experiment was run in triplicate. The 2 −∆∆CT method was used to calculate the relative mRNA expression levels [50].

### 4.10. Western Blot

After transfection, the culture medium was removed and the cells were washed twice with PBS. The cells were lysed with RIPA lysis buffer (Meilunbio, Dalian, China) containing PMSF protease inhibitors (Beyotime, Shanghai, China) on ice for 30 min, then centrifuged for 30 min. The cells were collected and bathed in ice for 30 min, then centrifuged for 30 min. Then, 10 μL of supernatant was extracted and diluted 10 times before being used to detect protein concentration using a BCA protein detection kit. The protein was mixed with Tris-Tricinesds-SDS-PAGE loading buffer (Solarbio, Beijing, China) and RAPI lysate, then denatured at 98 °C for 10 min. Protein samples were separated by SDS-PAGE, then transferred to the activated PVDF membranes (Merck Millipore, Darmstadt, Germany) by wet electrophoretic transfer. Subsequently, samples were blocked using 5% skim milk (Coolaber, Beijing, China) dissolved in TBST (Sangon biotech, Shanghai, China) for 2 h and subjected to TBST cleaning membranes. Then, PVDF membranes were incubated with primary antibody at 4 °C overnight. Membranes were washed with TBST and then incubated with a rabbit secondary antibody for 2 h at room temperature. Then, protein signals were visualized using the ECL reagent (Beyotime, Shanghai, China) and analyzed using ImageLab 6.1 software. The gray value of the target protein was divided by the gray value of β-actin, and this was used as the final statistical value [51].

### 4.11. Statistical analysis

All data are presented as means ± standard error from three independent experiments. The data were analyzed using Student’s *t*-test and SPSS 21.0 software (IBM, Armonk, New York, NY, USA). A value of *p* < 0.05 was declared statistically significant.

## 5. Conclusions

Our results revealed that *SDC3* upregulated the expression of *PPARG* through the *AMPK/SIRT1* signaling pathway to promote milk fat synthesis. It also regulated the inflammatory response of BMECs by regulating the activation of the *NF-κB* pathway—through the *AMPK/SIRT1* signaling pathway—decreasing the expression levels of inflammatory cytokines (*IL-6*, *IL-1β*, *TNF-α* and *COX-2*) and reducing ROS generation. *NF-κB p50* is an important target of *SDC3* in this process. Collectively, this study determined that *SDC3*, as a functional gene with complex biological effects, played a coregulatory role in milk lipid metabolism and inflammatory responses in BMECs. This study, therefore, could provide a theoretical foundation for future comprehensive breeding value evaluations of multi-gene effect functional genes in dairy cow molecular breeding.

## Figures and Tables

**Figure 1 ijms-24-06657-f001:**
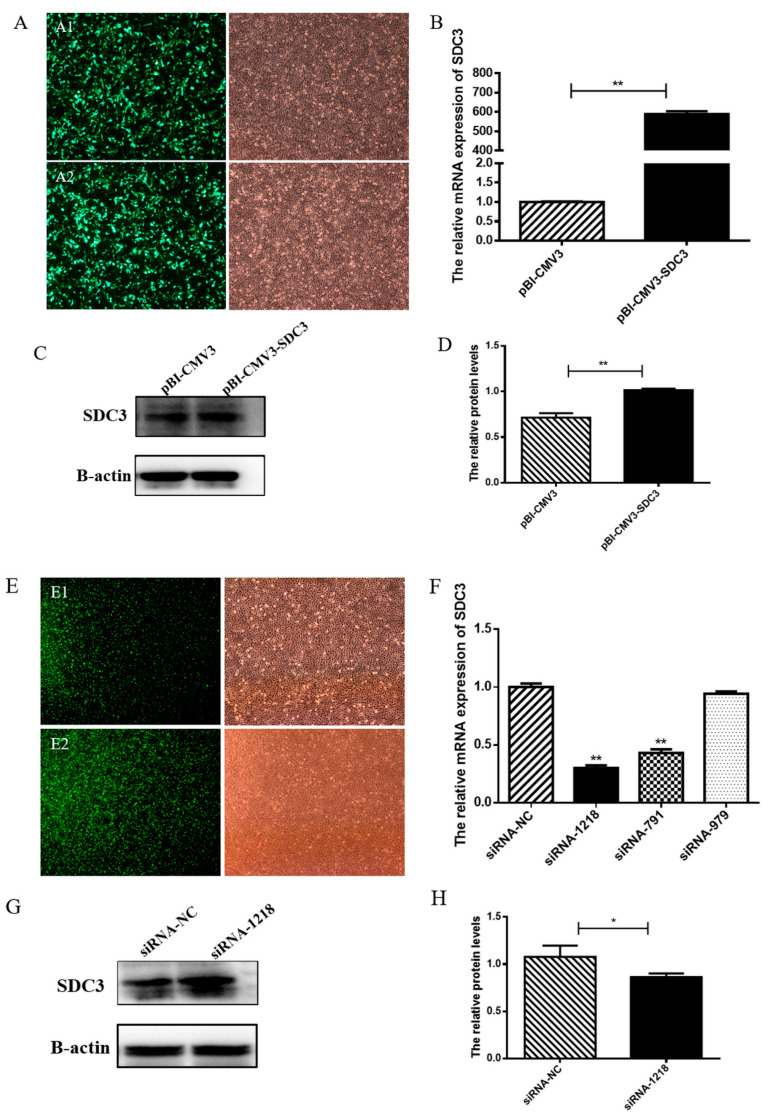
Identification of transfection efficiency of *SDC3* overexpression and *SDC3* knockout in bovine mammary epithelial cells (BMECs). The empty vector (pBI-CMV3) (**A1**), overexpression vector (pBI-CMV3-SDC3) (**A2**), small interference RNA (siRNA-1218) (**E1**), and its negative control (siRNA-NC) (**E2**) were transfected into BMECs (Magnification 40×). (**A**,**E**) Then, after 24 h, green fluorescence was observed under a fluorescence microscope (Magnification 40×). The expression of *SDC3* in BMECs was detected by RT-qPCR and Western blot. The overexpression of *SDC3* increases the transcription (**B**) and translation (**C**,**D**) of *SDC3* in BMECs, whereas the knockout of *SDC3* decreases the transcription (**F**) and translation (**G**,**H**) of *SDC3* in BMECs. Data presented as means ± SD (n = 3). Statistically significant differences are indicated: * *p* < 0.05, ** *p* < 0.01. pBI-CMV3, siRNA-NC = nonspecific scrambled control.

**Figure 2 ijms-24-06657-f002:**
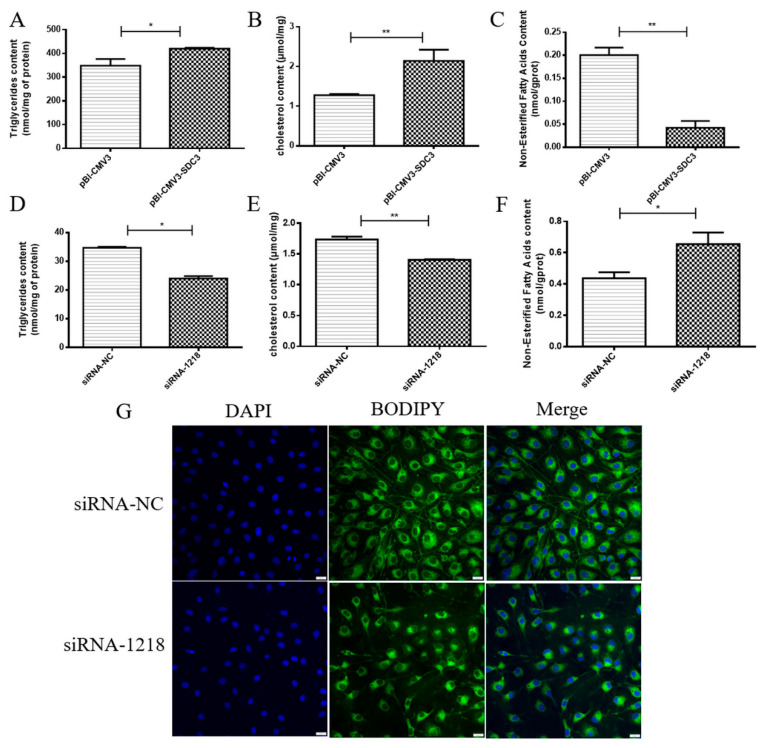
Effects of *SDC3* on milk fat synthesis. The vectors (pBI-CMV3, pBI-CMV3-SDC3) and siRNAs (siRNA-NC, siRNA-1218) were transfected into BMECs for 24 h. Triglyceride (**A**,**D**), cholesterol (**B**,**E**) and non-esterified fatty acid (**C**,**F**) kits were used to measure milk fat synthesis in BMECs, whereas BODIPY staining (**G**) was used to observe milk fat in BMECs. Data presented as means ± SD (n = 3). Statistically significant differences are indicated: * *p* < 0.05, ** *p* < 0.01. pBI-CMV3, siRNA-NC = nonspecific scrambled control. Scar bar = 20 μm.

**Figure 3 ijms-24-06657-f003:**
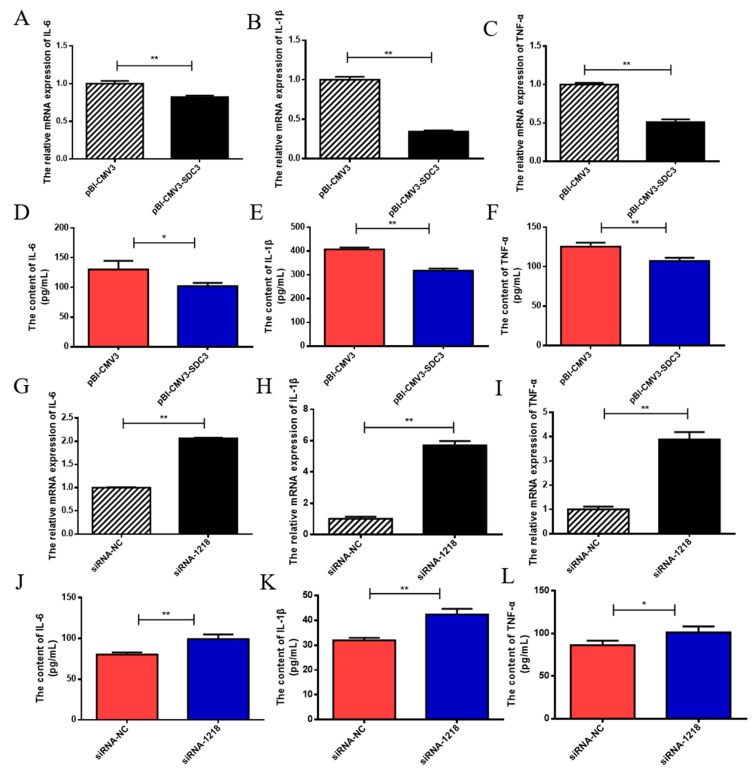
Effects of *SDC3* on the expression of inflammatory markers in BMECs. After BMECs were treated with vectors (pBI-CMV3, pBI-CMV3-SDC3) and siRNAs (siRNA-NC, siRNA-1218) for 24 h, the transcriptional levels of *IL-6, IL-1β*, *TNF-α* and *COX-2* in BMECs were detected by RT-qPCR (**A**–**C**,**G**–**I**,**M**,**N**). An ELISA kit was used to measure the protein expression levels of *IL-6, IL-1β*, and TNF-α in BMECs (**D**–**F**,**J**–**L**). The intracellular ROS level was measured using flow cytometry (**O**,**Q**), and ROS levels were analyzed (**P**,**R**). Data are presented as means ± SD (n = 3). Statistically significant differences are indicated: * *p* < 0.05, ** *p* < 0.01. pBI-CMV3, siRNA-NC = nonspecific scrambled control.

**Figure 4 ijms-24-06657-f004:**
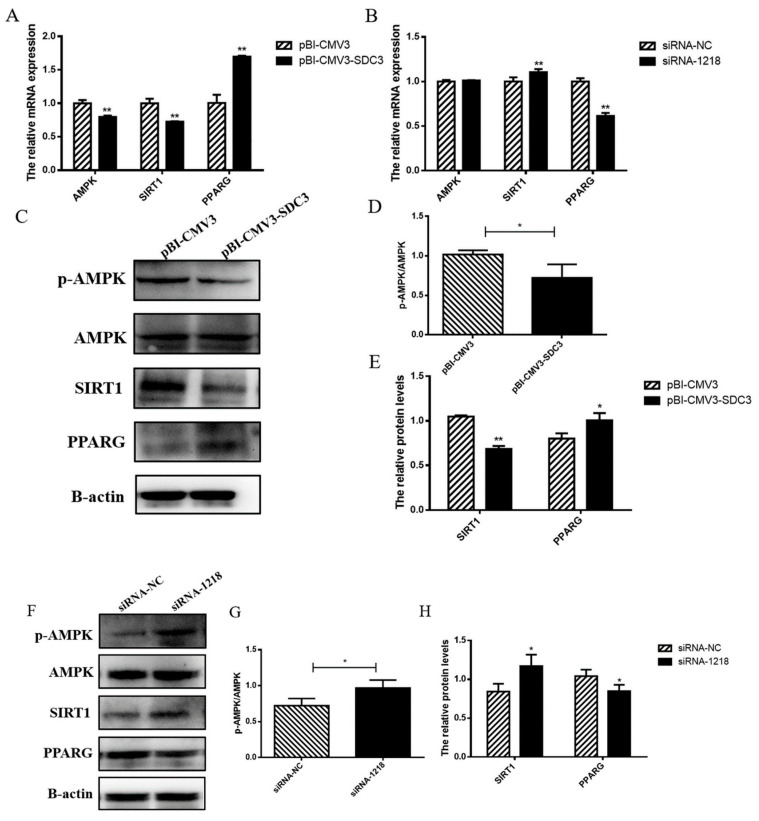
SDC3 regulates milk fat synthesis through the *AMPK* pathway. pBI-CMV3-SDC3 vector or siRNA-1218 transfected into BMECs. (**A**,**B**) The relative mRNA expression of *AMPK*, *SIRT1* and *PPARG* in BMECs, after over-expression and knockout of *SDC3*, as analyzed by RT-qPCR. (**C**,**F**) Western blotting, used to analyze the protein expression levels of Phosphorylated AMP-activated protein kinase (*P-AMPK*), *AMPK*, *SIRT1*, and *PPARG* in BMECs after overexpression and knockout of *SDC3*. (**D**,**G**) The ratios of phosphorylated-to-total *AMPK*, quantified by gray scale scan. (**E**,**H**) Relative folds of *SIRT1* and *PPARG* protein levels (protein/β-actin) from Western blots, quantified by gray scale scan. Data presented as means ± SD (n = 3). Statistically significant differences are indicated: * *p* < 0.05, ** *p* < 0.01. pBI-CMV3, siRNA-NC = nonspecific scrambled control.

**Figure 5 ijms-24-06657-f005:**
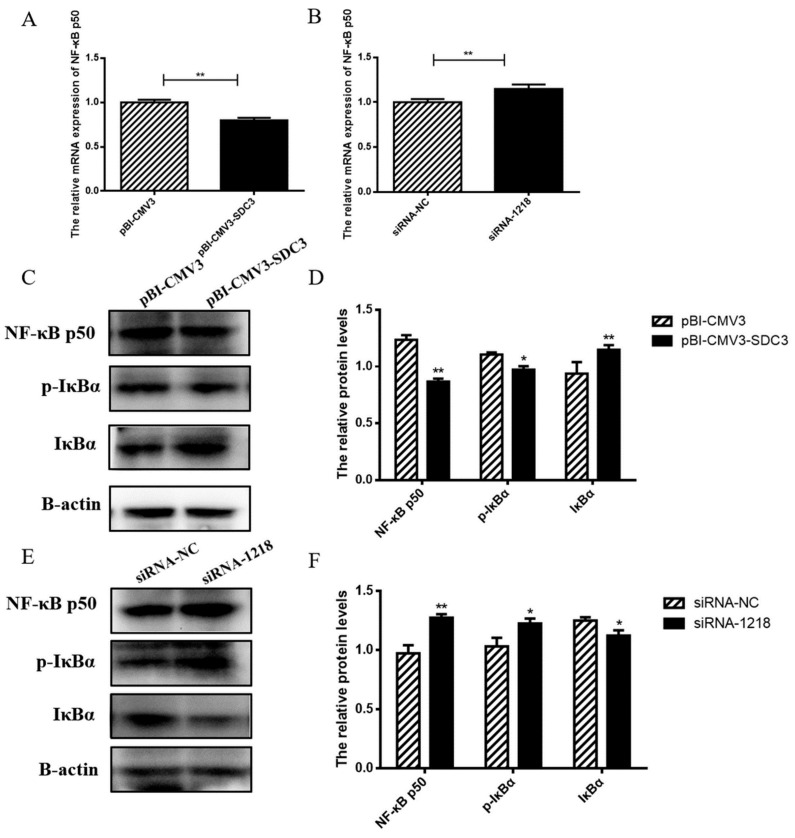
*SDC3* is a negative regulator of the *NF-κB* signaling pathway. pBI-CMV3-SDC3 vector or siRNA-1218, was transfected into BMECs. (**A**,**B**) RT-qPCR was used to detect the transcriptional levels of *NF-κB p50*. (**C**,**E**) The expression levels of *NF-κB p50*, Phospho-IκB Alpha (*P-IκBα*), and *IκBα* in all groups of BMECs were examined using Western blot. (**D**,**F**) Relative folds of *NF-κB p50*, *P-IκBα*, and *IκBα* protein levels (protein/β-actin) from the Western blots, quantified by gray scale scan. Data are presented as means ± SD (n = 3). Statistically significant differences are indicated: * *p* < 0.05, ** *p* < 0.01. pBI-CMV3, siRNA-NC = nonspecific scrambled control.

**Figure 6 ijms-24-06657-f006:**
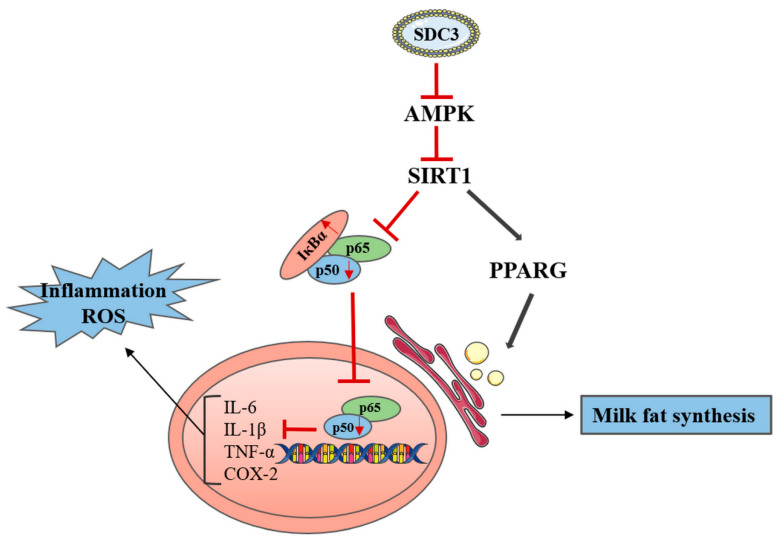
Summary of the mechanism by which *SDC3* cooperates with the regulation of milk fat synthesis and inflammatory response. The overexpression of *SDC3* can promote the synthesis of milk fat in BMECs by inhibiting the activation of *AMPK*, downregulating the expression of downstream gene SIRT1, and upregulating the expression of *PPARG*. In addition, the overexpression of *SDC3* inhibits the expression of *NF-κB p50* and *P-IκBα*, increases the expression of *IκBα*, and inhibits the *NF-κB* pathway, reducing the expression of downstream inflammatory markers (*IL-6*, *IL-1β*, *TNF-α*, and *COX-2*) and ROS generation—and, finally, inhibiting the inflammatory response of BMECs.

**Table 1 ijms-24-06657-t001:** Primer sequences used for quantitative real-time PCR ^1^.

Gene	Sequence Number	Primer Sequence (5′ to 3′)	Product Size (bp)
** *AMPKα1* **	XM_005221557.3	F: CCCGTATTATTTGCGTGTTCG	160
		R: CTGTGGCGTAGCAGTCCCT	
** *SIRT1* **	NM_001192980.3	F: TGAAGAATGCTGTCTCCAAT	271
		R: GCGTTTACTAATCTGCTCCT	
** *PPARG* **	NM_181024.2	F: TGACCCGATGGTTGCAGATTAT	100
		R: ATGAGGGAGTTGGAAGGCTCT	
** *NF-* ** ** *κ* ** ** *B p50* **	NM_001076409.1	F: AACATCCACCTGCATGCACAC	155
		R: GGCATCTGTCATTCGTGCTTC	
** *IL-6* **	NM_173923.2	F: GATGCTTCCAATCTGGGTTCA	115
		R: TCCTGATTTCCCTCATACTCG	
** *IL-1β* **	NM_174093.1	F: GATGGCTTACTACAGTGACGA	254
		R: AGATGAATGAAAGGATGCTC	
** *TNF-* ** ** *α* **	XM_005223596.4	F: ACGGGCTTTACCTCATCTACTC	132
		R: TGGCAGACAGGATGTTGACC	
** *COX-2* **	YP_209208.1	F: CCAGAGCTCTTCCTCCTGTG	213
		R: AAGCTGGTCCTCGTTCAAAA	
** *SDC3* **	NM_001206531.2	F: CGCCCTCTTTGCTGCCTT	103
		R: TGTGACGCTCGCCTGCTT	
** *ACTB* **	NM_173979.3	F:AGAGCAAGAGAGGCATCC	133
		R:TCGTTGTAGAAGGTGTGGT	

^1^ F, forward, R, reverse; *AMPKα1*, protein kinase AMP-activated catalytic subunit alpha 1; *SIRT1*, sirtuin 1; *PPARG*, peroxisome proliferator activated receptor gamma; *NF-κB p50*, nuclear factor kappa B subunit 1; *IL-6*, interleukin 6; *IL-1β*, interleukin 1 beta; *TNF-α*, tumor necrosis factor; *COX-2*, cytochrome c oxidase subunit II; *SDC3*, Syndecan-3; and *ACTB*, actin beta.

## Data Availability

All data generated or analyzed during this study are available from the corresponding author by request.

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
