# Peer review of "Syndecan-3 Coregulates Milk Fat Metabolism and Inflammatory Reactions in Bovine Mammary Epithelial Cells through AMPK/SIRT1 Signaling Pathway"

_ijms, 2023, doi:10.3390/ijms24076657_

Round 1

Reviewer 1 Report

Reviewer Comments to Authors:

The authors present original research that adds useful information to those who are about to implement molecular tools in breeding programs for dairy cattle. However, the manuscript needs some minor revisions:

1.      The title: “AMPK/SIRT1 signaling pathway” instead of “AMPK/SIRT1 signal pathway” would have been more accurate

2.      There are many grammatical and typing errors throughout the manuscript that need to be revised. Examples:

·         Abstract, line 29: Please change proliferator activated to "proliferator-activated.

·         Abstract, line 34: Please change inhibits to inhibiting.

·         Abstract, line 34: Please change NF-kB to the NF-kB.

·         Abstract, lines 41, 42 and 43: Please rewrite the sentence to make it more comprehensible.

·         Background, line 57: Please change We to we.

·         Background, line 60: Please change suggest to suggests.

·         Background, line 71: Please fix the comma error.

·         Background, line 76: Please change bacterial to the bacterial.

·         Background, line 84: Please delete “which is” from the sentence, to make it more concise.

·         Background, line 88: Please change “cloned in rat” to “cloned in the rat”.

·         Background, line 104: Please change trans-10, cis-12 (t10c12-CLA) to trans-10, cis-12 conjugated linoleic acid (t10c12-CLA).

·         Background, line 107: Please change phosphorylation α-subunit Thr site to phosphorylation of α-subunit Thr site.

·         Background, line 127: Please change “coregulates” to “coregulation”.

·         Methods, line 135: Please change Nanjing Jian cheng to Nanjing Jiancheng.

·         Methods, line 146: Did you mean:taken to the Molecular genetics laboratory, College of animal science, Jilin University

·         Methods, line 156:  Please delete “which”

·         Methods, line 164:  Please add cultured.

·         Methods, lines 198, 199, 200, 201, 202:  Please rewrite the sentence to make it grammatically correct.

·         Methods, line 234:  Please change primer to primers.

·         Methods, line 235:  Please change “under” to “in”.

·         Methods, line 256:  Please change “signal” to “signals”.

·         Results, line 269:  Did you mean: the vectors and siRNAs were transfected into cells?

·         Results, line 270: Perhaps “analyzed” instead of „detected“ would be more appropriate.

·         Results, line 274: Please change “transcriptional” to “transcription”.

·         Results, line 274: Please change: “siRNA-1218 group obvious decreased… but obvious increased the content of NEFA” to “the siRNA-1218 group obviously decreased… but obviously increased the content of NEFA”.

·         Results, line 306: Please change: “and play a positive role in regulating milk fat synthesis, but also promote the metabolism of NEFA” to “and plays a positive role in regulating milk fat synthesis, but also promotes the metabolism of NEFA”.

·         Discussion, line 481: Please change: dairy to the dairy.

·         Discussion, line 492: Please change “indicating” to “indicates”.

·         Discussion, line 503: Please change “inflammatory” to “the inflammatory”.

·         Discussion, line 505: Please change: “it was named because” to “so named because”.

·         Discussion, lines 528, 529 and 530: Please rewrite the sentence to make it grammatically correct.

·         Discussion, lines 534 and 535: Perhaps All in all, from our results we can conclude that SDC3 regulates the activation of NF-kB signal pathway through NF-kB P50, thus exerting its anti-inflammatory activity. would be better.

·         Conclusions, lines 538, 539, 540, 541 and 542: This sentence should be reviewed and edited to be more concise and to better reflect the results. Perhaps:In summary, our results revealed that SDC3 upregulates the expression of PPARG through AMPK/SIRT1 signalling pathway to promote milk fat synthesis, and also regulates the inflammatory response of BMECs by regulating the activation of NF-kB pathway through AMPK/SIRT1 signalling pathway to decrease the expression of inflammatory cytokines (IL-6, IL-1β, TNF-α and COX-2) and ROS generation. would be more appropriate.

3.      Some Figure legends aren't adequately written.

·         Figure 3: Did you mean: the transcriptional levels of IL-6, IL-1 β and TNF- α in BMECs were detected by RT-qPCR (A, B, C, G, H, I, M and N), and ELISA kit was used to measure the protein expression levels of IL-6, IL- 1 β and TNF-α in BMECs (D, E, F, J, K and L). The intracellular ROS level was measured using flow 349 cytometry (O, Q), and ROS levels were analyzed (P,R).?

·         Figure 4. Did you mean: The relative mRNA expression of AMPK, SIRT1 and PPARG in BMECs after over-expression and knockout of SDC3 was analyzed by RT-qPCR. (A and B) Western blotting was used to analyze the protein expression of P-AMPK, AMPK, SIRT1 and PPARG in BMECs after over-expression and knockout of SDC3. (C, F) The ratios of phosphorylated to total AMPK were quantified by gray scale scan. (D and G) Relative folds of SIRT1 and PPARG protein levels (protein/β-actin) from the western blots were quantified by gray scale scan.(E, H)?

·         Figure 5. Did you mean: “RT-qPCR was used to detect the transcriptional level of NF-kB P50. (A and B) The expression of NF-kB P50, P-IkB α and IkB α in all groups of BMECs were examined using Western blot. (C and E) Relative folds of NF-kB P50, P-IkB α and IkB α protein levels (protein/β-actin) from the western blots were quantified by gray scale scan (D and F).”

Author Response

Dear Reviewer,

On behalf of all the authors of the submitted manuscript “SDC3 coregulates milk fat metabolism and inflammatory reaction in Bovine Mammary Epithelial Cells through AMPK/SIRT1 signaling pathway” (ID: ijms-2209697), we thank you so much for your kind reviews and meaningful suggestions. We have studied comments carefully and have made corrections which should meet with approvals we hope. The main revision in the paper (highlight changes in red) and the responses to the reviewers’ comments are as following:

Responds to Reviewer 1:

There are many grammatical and typing errors throughout the manuscript that need to be revised.

Response: Thank you very much for your advice. Your suggestion is very useful to us. We have corrected the grammatical and typing errors you pointed out in the manuscript and used English editing services to correct the grammatical and spelling errors in the manuscript. In addition, we have made some changes according to the opinions of reviewer 2, and some of the contents have been deleted, so some of the grammatical and spelling mistakes you have pointed out do not correspond to the manuscript. (See lines 27, 28, 24, 33, 34, 48, 51, 63, 67, 101, 105, 124, 132, 143, 155, 163, 233, 234, 258, 271, 272, 276, 306, 489, 499, 511, 513).

  1. The title: “AMPK/SIRT1 signaling pathway” instead of “AMPK/SIRT1 signal pathway” would have been more accurate.

Response: Thank you very much for your suggestion. Your suggestions have been very useful to us. We have corrected this error as you suggested (see line 4).

  1. Methods, lines 198, 199, 200, 201, 202: Please rewrite the sentence to make it grammatically correct.

Response: Thank you very much for your suggestion. Your suggestions have been very useful to us. We have corrected this error as you suggested (see line 195-200).

  1. Discussion, lines 528, 529 and 530: Please rewrite the sentence to make it grammatically correct.

Response: Thank you very much for your suggestion. Your suggestions have been very useful to us. We have corrected this error as you suggested (see line 537-539).

  1. Discussion, lines 534 and 535: Perhaps “All in all, from our results we can conclude that SDC3 regulates the activation of NF-kB signal pathway through NF-kB P50, thus exerting its anti-inflammatory activity.” would be better.

Response: Thank you very much for your suggestion. Your suggestions have been very useful to us. We have corrected this error as you suggested (see line 543-545).

  1. Conclusions, lines 538, 539, 540, 541 and 542: This sentence should be reviewed and edited to be more concise and to better reflect the results. Perhaps:“In summary, our results revealed that SDC3 upregulates the expression of PPARG through AMPK/SIRT1 signalling pathway to promote milk fat synthesis, and also regulates the inflammatory response of BMECs by regulating the activation of NF-kB pathway through AMPK/SIRT1 signaling pathway to decrease the expression of inflammatory cytokines (IL-6, IL-1β, TNF-α and COX-2) and ROS generation. “would be more appropriate.

Response: Thank you very much for your suggestion. Your suggestions have been very useful to us. We have corrected this error as you suggested (see line 547-552).

  1. Some Figure legends aren't adequately written.

Response: Thank you very much for your suggestion. Your suggestions have been very useful to us. We have checked the Figure legends of Figure 3, 4 and 5 and made corrections (see line 350-355, 390-396, 429-433).

Reviewer 2 Report

The present manuscript discussed the role of syndecan-3 (SDC3) in milk fat metabolism and inflammatory processes in bovine mammary epithelial cells (BMECs). Overall, this is original study that brings novelty in this research field with some significant data. SDC3 was shown to exert anti-inflammatory effects (besides effects on milk fat metabolism) and one common cellular signaling pathway in these processes was identified - AMPK/SIRT1/PPARG.

However, the manuscript is not well written, some parts are redundant, some parts are confusing and there are too many linguistic and technical errors in the manuscript. The whole manuscript needs to be thoroughly checked and modified.

First, please modify the font theme and size in the whole manuscript to be uniform.

In the Title: please define both SDC3 and BMEC (it is not so well known for the general scientific community), first letter in the term ‘inflammatory’ does not need to be capital.

In the Abstract: ‘non-esterified fatty acids’ instead of ‘Non-Esterified Fatty Acids’, ‘kinase’ instead of ‘kinas’…

Introduction needs to be substantially rearranged. It should be focused on SDC3. The part about PPARα is not so relevant for this specific topic. Besides, the questions at the lines 66-68 I find redundant, it should be formulated in a different way at the end of the Introduction, when defining objectives of the study. There are also some technical errors - ‘we’ instead of ‘We’ (line 57), ‘sodium butyrate’ instead of ‘Sodium Butyrate’ etc. Besides, why do you find that study about SB (ref 29) relevant since it is well known that NF-κB is a central mediator of inflammation?

In Methods, instead of RT-qPCR and ELISA, the subtitles should be Gene/protein expression analysis, qPCR and ELISA are just techniques for these investigations. In qPCR description and table 1 with primer sequences, reference gene (ACTB) is missing.

In Results, Figure 3 should be more compact (on one page).

Discussion and Conclusion are generally well described (however, some technical issues are present in this part too and require to be thoroughly checked).

Author Response

Dear Reviewer,

On behalf of all the authors of the submitted manuscript “SDC3 coregulates milk fat metabolism and inflammatory reaction in Bovine Mammary Epithelial Cells through AMPK/SIRT1 signaling pathway” (ID: ijms-2209697), we thank you so much for your kind reviews and meaningful suggestions. We have studied comments carefully and have made corrections which should meet with approvals we hope. The main revision in the paper (highlight changes in red) and the responses to the reviewers’ comments are as following:

Responds to Reviewer 2:

  1. First, please modify the font theme and size in the whole manuscript to be uniform.

Response: Thank you very much for your suggestion. Your suggestions have been very useful to us. We have reviewed and modified the font theme and size in the manuscript.

  1. In the Title: please define both SDC3 and BMEC (it is not so well known for the general scientific community), first letter in the term ‘inflammatory’ does not need to be capital.

Response: Thank you very much for your suggestion. We have defined SDC3 and BMECs in the title and modified the term ‘inflammatory’ (see line 1-4).

  1. In the Abstract: ‘non-esterified fatty acids’ instead of ‘Non-Esterified Fatty Acids’, ‘kinase’ instead of ‘kinas’…

Response: Thank you very much for your suggestion. We have corrected the two errors you pointed out in the Abstract (see line 24).

  1. Introduction needs to be substantially rearranged. It should be focused on SDC3. The part about PPARα is not so relevant for this specific topic. Besides, the questions at the lines 66-68 I find redundant, it should be formulated in a different way at the end of the Introduction, when defining objectives of the study. There are also some technical errors - ‘we’ instead of ‘We’ (line 57), ‘sodium butyrate’ instead of ‘Sodium Butyrate’ etc. Besides, why do you find that study about SB (ref 29) relevant since it is well known that NF-κB is a central mediator of inflammation?

Response: Thank you very much for your suggestion. Your suggestions have been very useful to us. We have rearranged the introduction, modified the description in the SDC3 section of the introduction (see line 73-98), deleted the PPARα section, modified the questions section of lines 66-68 (see line 60, 61), modified the technical errors you pointed out (see line 48), and deleted the references to SB (ref29) (see line114). We appreciate your suggestions for the introduction section. If you have any suggestions for the revised introduction, we welcome your reply.

  1. In Methods, instead of RT-qPCR and ELISA, the subtitles should be Gene/protein expression analysis, qPCR and ELISA are just techniques for these investigations. In qPCR description and table 1 with primer sequences, reference gene (ACTB) is missing.

Response: Thank you very much for your suggestion. We have modified the subtitles of the method regarding RT-qPCR and ELISA (see line 203, 255), and added the primer sequence of the reference gene (ACTB) to table 1 (see line 238, Table 1).

  1. In Results, Figure 3 should be more compact (on one page).

Response: Thank you very much for your suggestion. We have modified Figure 3 and recomposed the original three images into two to make them more compact (see line 344).

  1. Discussion and Conclusion are generally well described (however, some technical issues are present in this part too and require to be thoroughly checked).

Response: Thank you very much for your suggestion. We have used an English editing service to correct all grammatical and spelling errors in the manuscript.

Round 2

Reviewer 2 Report

The manuscript has been much improved in comparison to the previous version and therefore I support it to be published in this form.